

# Co-inoculation of mycorrhizal fungi and plant growth-promoting rhizobacteria improve growth, biochemical and physiological attributes in *Dracocephalum kotschyi* Boiss. under water deficit stress

Saeid Gasemi[1], Hassan Mahdavikia[2], Esmaeil Rezaei-Chiyaneh[3], Farzad Banaei-Asl[4], Aria Dolatabadian[5] and Amir Sadeghpour[6]

[1] Department of Medicinal Plants, Urmia University, Miandoab, Urmia, Iran
[2] Department of Medicinal Plants and Horticulture, Shahid Bakeri Higher Education Center of Miandoab, Urmia University, Urmia, Iran
[3] Department of Plant Production and Genetics, Urmia University, Urmia, Iran
[4] Biotechnology Research Department, Research Institute of Forests and Rangelands, Agricultural Research, Education and Extension Organization, Tehran, Iran
[5] School of Biological Sciences, The University of Western Australia, Perth, Western Australia, Australia
[6] School of Agricultural Sciences, Southern Illinois University, Carbondale, IL, United States of America

Corresponding author
Hassan Mahdavikia,
h.mahdavikia@urmia.ac.ir,
hassanmahdavikia.210@gmail.com

## ABSTRACT

**Background.** Because of swift climate change, drought is a primary environmental factor that substantially diminishes plant productivity. Furthermore, the increased use of chemical fertilizers has given rise to numerous environmental problems and health risks. Presently, there is a transition towards biofertilizers to enhance crops' yield, encompassing medicinal and aromatic varieties.

**Methods.** This study aimed to explore the impacts of plant growth-promoting rhizobacteria (PGPR), both independently and in conjunction with arbuscular mycorrhizal fungi (AMF), on various morphological, physiological, and phytochemical characteristics of *Dracocephalum kotschyi* Boiss. This experimentation took place under different irrigation conditions. The irrigation schemes encompassed well watering (WW), mild water stress (MWS), and severe water stress (SWS). The study evaluated the effects of various biofertilizers, including AMF, PGPR, and the combined application of both AMF and PGPR (AMF + PGPR), compared to a control group where no biofertilizers were applied.

**Results.** The findings of the study revealed that under water-stress conditions, the dry yield and relative water content of *D. kotschyi* Boiss. experienced a decline. However, the application of AMF, PGPR, and AMF + PGPR led to an enhancement in dry yield and relative water content compared to the control group. Among the treatments, the co-application of AMF and PGPR in plants subjected to well watering (WW) exhibited the tallest growth (65 cm), the highest leaf count (187), and the most elevated chlorophyll *a* (0.59 mg g$^{-1}$ fw) and *b* (0.24 mg g$^{-1}$ fw) content. Regarding essential oil production, the maximum content (1.29%) and yield (0.13 g plant $^{-1}$) were obtained from mild water stress (MWS) treatment. The co-application of AMF and PGPR resulted in the highest essential oil content and yield (1.31% and 0.15 g plant$^{-1}$, respectively). The analysis of *D. kotschyi* Boiss. essential oil identified twenty-six compounds, with major

constituents including geranyl acetate (11.4–18.88%), alpha-pinene (9.33–15.08%), Bis (2-Ethylhexyl) phthalate (8.43-12.8%), neral (6.80–9.32%), geranial (9.23–11.91%), and limonene (5.56–9.12%). Notably, the highest content of geranyl acetate, geranial, limonene, and alpha-pinene was observed in plants subjected to MWS treatment following AMF + PGPR application. Furthermore, the co-application of AMF, PGPR, and severe water stress (SWS) notably increased the total soluble sugar (TSS) and proline content. In conclusion, the results indicate that the combined application of AMF and PGPR can effectively enhance the quantity and quality of essential oil in *D. kotschyi* Boiss., particularly when the plants are exposed to water deficit stress conditions.

# INTRODUCTION

Iran receives approximately two-thirds less rainfall than the global average (*Ostadi et al., 2023*). Over 88% of Iran's farmlands are in arid and semi-arid regions (*Vaghefi et al., 2019*). Water scarcity conditions, or drought stress, have a detrimental impact on plant functioning. This includes a decrease in root growth and nutrient absorption, a reduction in photosynthesis capacity, and an increase in the content of reactive oxygen species (ROS) and lipid peroxidation within the plant's membranes (*Seleiman et al., 2021*; *Rezaei-Chiyaneh et al., 2018*). Plants' reaction to water deficit stress is contingent upon its severity and duration, leading to a decrease in plant productivity ranging from 13% to 94% (*Ostadi et al., 2022*).

Furthermore, insufficient soil moisture diminishes the activity of the microbial population (*Zamani et al., 2023*). This decrease hampers the accessibility and solubility of nutrients for plants (*Rezaei-Chiyaneh et al., 2021a*). Moreover, the increased utilization of chemical fertilizers to attain peak crop yields amplifies production expenses, elevates the risk of environmental pollution, and poses threats to both human and animal well-being (*Ganbari Torkamany et al., 2023*). Utilizing biofertilizers such as arbuscular mycorrhizal fungi (AMF) and plant growth-promoting rhizobacteria (PGPR) to decrease or even eliminate chemical input usage presents a commendable resolution to address these apprehensions within the realm of sustainable agriculture (*Taghizadeh et al., 2023*). Biofertilizers contain beneficial bacteria and fungi that convert essential nutrients from inaccessible to accessible forms during biological processes (*Hafez, Popov & Rashad, 2021*; *Heydarzadeh et al., 2022*). Studies have indicated that PGPR can stimulate plant development by producing diverse compounds. These bacteria assist in element absorption, atmospheric nitrogen stabilization, mineral solubilization like phosphate, and the synthesis of plant hormones such as auxins and gibberellins. These activities collectively enhance plant performance (*Vocciante et al., 2022*).

Furthermore, PGPR can alleviate the adverse effects of stressful conditions such as drought and salinity. They achieve this by modifying root morphology, augmenting the

buildup of osmolytes within plant cells, and enhancing water uptake mechanisms (*Ahmad et al., 2022*). Inoculating plant roots with AMF presents another auspicious and productive approach to enhancing plant performance and resilience under drought-stress conditions. The symbiotic relationship between AMF and over 80% of plant species has a constructive impact on plant growth. It achieves this by augmenting the uptake of inorganic nutrients and water, elevating the rate of photosynthesis, and fostering the production of both primary and secondary metabolites. These metabolites are pivotal in bolstering plants' ability to withstand and endure stressful conditions (*Amani Machiani et al., 2022*; *Amiri, Nikbakht & Etemadi, 2015*). One of the viable strategies for achieving thriving agriculture in arid and semi-arid regions involves cultivating drought-resistant plants. Medicinal and aromatic plants (MAPs) cultivation has gained prominence in these challenging environments due to their remarkable resilience to adverse conditions. Despite stress, MAPs can sustain their performance by synthesizing secondary metabolites, including alkaloids and phenolic compounds (*Amani Machiani et al., 2022*). *Heydarzadeh et al. (2023b)* indicated that water deficit stress decreased chlorophyll content, relative water content, grain yield, biological yield, and essential oil yield of dragon head compared to the control. *Alipour et al. (2021)* showed that applying AMF and PGPR while reducing the adverse effects of drought stress by providing nutrients of N, P, and K improves the quality of fennel essential oil. *Eshaghi Gorgi et al. (2022)* stated that the application of AMF and PGPR on *Melissa officinalis* L. plants could alleviate the adverse effects of water scarcity by boosting leaf water potential, the efficiency of carbon dioxide use, transpiration rate, nutrient availability, water supply to roots. Moreover, *Melissa officinalis* L. plants' growth, development, and dry weight yield can also improve under water deficit stress.

Dragon head (*D. kotschyi* Boiss.) is a member of the Lamiaceae family and holds the distinction of being an endemic and endangered plant species native to Iran (*Poursalavati, Rashidi-Monfared & Ebrahimi, 2021*). This plant possesses antispasmodic, antibacterial, and analgesic properties, making it extensively utilized for addressing issues such as pain, seizures, fever, inflammation, rheumatic pain, and wound healing (*Ghavam et al., 2021*; *Sadraei, Asghari & Kasiri, 2015*). It has been reported that the main constituents of essential oil (EO) in *D. kotschyi* Boiss. differ due to different environmental conditions but mainly include $\alpha$-pinene, limonene, $\beta$-bourbonene, geranial, terpinene-4-ol, *etc.* (*Sonboli, Mirzania & Gholipour, 2019*). Limonene, one of the main components of *D. kotschyi* Boiss. EO is effective as an anti-tumour, antiviral, bactericidal, cancer prevention agent, anti-candida, expectorant, fungus growth inhibitor, anti-spasm and pain reliever (*Moradi, Ghavam & Tavili, 2020*).

Furthermore, beyond diminishing the effectiveness of chemical fertilizers under water scarcity conditions, the extensive application of these fertilizers also adversely impacts the bioactive compounds in MAPs (*Asghari et al., 2023*). Hence, substituting chemical fertilizers with environmentally friendly biofertilizers to enhance nutrient uptake and augment plant resilience to water stress becomes imperative. Moreover, the impact of biofertilizers on perennial plants, such as cumin, remains inadequately investigated under drought stress circumstances, and there exists a dearth of substantial information in this domain (*Rezaei-Chiyaneh et al., 2021b*).

Consequently, the primary objective of this study was to delve into the influence of AMF and plant PGPR on the agronomic traits, dry yield, and the phytochemical and physiological characteristics of *D. kotschyi* Boiss. cultivated under conditions of water scarcity. Our working hypotheses were as follows: (i) the combined application of AMF and PGPR would alleviate the adverse effects caused by water deficit stress; (ii) the introduction of AMF and PGPR would enhance both the quantity and quality of essential oil production under water deficit stress conditions; and (iii) the inoculation of AMF and PGPR would contribute to improved growth, biochemical attributes, and physiological responses in *D. kotschyi* Boiss. when confronted with water deficit stress.

## MATERIALS & METHODS

### Experimental design and treatments

The study was undertaken during the 2019 growing season within a controlled environment at Shahid Bakri Miandoab, situated in West Azerbaijan, Iran (45°10′E, 37°44′N, and an elevation of 1,338 m above sea level). The primary objective of this investigation was to evaluate the effects of distinct irrigation regimes and biofertilizers on the morphological, physiological, and phytochemical attributes of *Dorema kotschyi* Boiss. The experimental design employed a factorial approach within a randomized complete block arrangement, with each treatment combination being replicated three times. The initial factor under investigation pertained to distinct irrigation regimes, namely optimal irrigation, denoted watering at 100% field capacity (WW) to maintain moisture without inducing stress; moderate water stress involving irrigation at 80% field capacity (MWS); and pronounced water stress entailing irrigation at 60% field capacity (SWS). The second factor centered on the administration of biofertilizers, encompassing a control group devoid of biofertilizer application, the introduction of arbuscular mycorrhizal fungi (AMF), plant growth-promoting rhizobacteria (PGPR), and a concurrent application of both AMF and PGPR. Table 1 presents a comprehensive summary of the chemical and physical attributes of the soil, analyzed at a depth ranging from 0 to 30 cm. Meanwhile, Figure 1 depicts the average monthly rainfall and air temperature measurements.

The experimental arrangement entailed the utilization of pots with dimensions measuring $80 \times 25 \times 25$ cm, each uniformly filled with soil to an equivalent level. Ten plants were cultivated within each pot, and the planting commenced in mid-March. The plant growth-promoting rhizobacteria (PGPR), acquired from Green Biotech Ltd., recognized as Zist Fanavar Sabz in Persian, constituted a composite comprising phosphate-solubilizing bacteria (*Bacillus lentus* and *Pseudomonas putida*) at a concentration of 109 active bacteria per gram, nitrogen-fixing bacteria (*Azotobacter*) at a concentration of 109 active bacteria per gram, and potassium-solubilizing bacteria (*Pseudomonas genus*) at a concentration of 108 active bacteria per gram. The contents of the packets were blended with sterile distilled water and subsequently applied as a spray onto the seeds of *Dorema kotschyi* Boiss. This procedure ensured the establishment of a consistent coating on the seed surface. The biofertilizer containing AMF, specifically Myco-root℗ sourced from Zist Fanavar Sabz Company in Iran, encompassed a trio of species: *Glomus mossea*, *Glomus etunicatum*,

**Table 1  Soil chemical properties.**

| Texture | pH | EC (dS m$^{-1}$) | Organic matter (%) | Total N (%) | Phosphorus (mg kg$^{-1}$) | Potassium (mg kg$^{-1}$) |
|---------|-----|------------------|--------------------|-------------|----------------------------|---------------------------|
| Silty | 7.85 | 0.91 | 0.87 | 0.07 | 10.76 | 178.35 |

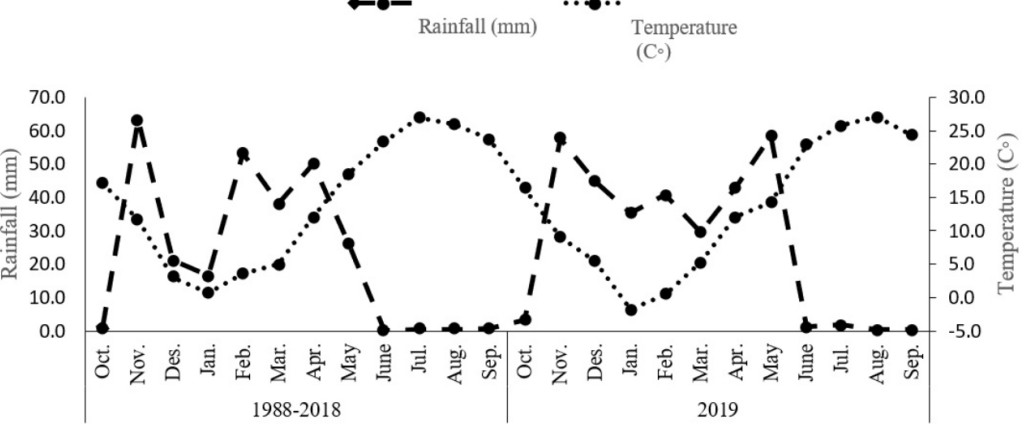

**Figure 1  Total rainfall and average monthly air temperature for the 1988–2018.**

and *Glomus intraradices*. This study introduced the AMF biofertilizer into the designated treatments by incorporating it into the soil beneath the planted seedlings. For each pot, 50 grams of the AMF treatment were meticulously combined with the soil. The cultivation area was divided into six rows, each spaced at a 10 cm interval. Subsequently, the seedlings were positioned within the same row, maintaining a separation of 10 cm between each seedling, and were planted at a depth of three cm.

## Observation

### *Agronomic traits and dry yield*

During the stage of 50% flowering, a random selection of five plants was made from each pot to evaluate yield components and agronomic characteristics. The above-ground portion of the *Dorema kotschyi* Boiss. The plant was harvested and dried in a forced-air oven set at 68 °C for 72 h. Following a two-day interval, the dried plant material was weighed to ascertain the dry yield measure.

### *Essential oil extraction and analysis*

The *D. kotschyi* Boiss. EO was extracted from 40 g dried aerial parts, ground and inserted into 1 L glass flasks filled with 300 mL of distilled water. Then, they were boiled for 3 h to exhaust the whole plant material. The EO was collected in specific glass containers after adding sodium sulphate and kept at 4 °C in the darkness until GC-FID and GC–MS analyses (*Mohammadzadeh et al., 2022*). Also, the EO yield, as g plant$^{-1}$, was calculated by multiplying the dry yield with EO content. After extraction of EO, the required amount of

sodium sulphate was added, and samples were kept at 4 °C in darkness for further chemical analysis.

Moreover, the EO constituents of *D. kotschyi* Boiss. were analyzed using GC-MS (7890/5975A GC/MSD; Agilent, Santa Clara, CA, USA) and GC-FID (7990B; Agilent, Santa Clara, CA, USA). For GC–MS analysis, an Agilent 7890A-5975C gas chromatograph manufactured in the United States was employed. This chromatograph was equipped with an HP-5MS capillary column comprising a 5% phenyl methylpolysiloxane composition, with dimensions of 30 m in length, an inner diameter of 0.25 mm, and a film thickness of 0.25 μm. The oven temperature was programmed as follows: an initial duration of 3 min at 80 °C, followed by a subsequent increase of 8 °C per minute until reaching 180 °C, which was then maintained for 10 min. The transfer line temperature was set at 240 °C. The carrier gas employed was helium, flowing at 1 ml per minute. The injector employed a split ratio of 1:50, and mass range acquisition occurred within the range of 40 to 500 m/z in electron impact (EI) mode at an energy of 70 electron volts (eV). The essential oil constituents (EO) were identified per the procedure outlined by *Morshedloo et al. (2018)*. In brief, the components' retention indices (RIs) were determined by comparing them with a standard mixture of n-alkanes (C7-C28) from Sigma-Aldrich (St. Louis, MO, USA). The calculated RIs and the corresponding mass spectra were compared against entries within commercial libraries and existing literature sources (*Adams, 2007*; NIST 08, 2008). To aid in identifying the major components, authentic standards available from Sigma-Aldrich were also co-injected. An Agilent 7890, an instrument produced by Agilent Technologies (Santa Clara, CA, USA) was utilized for the GC-FID analysis. This instrument was coupled with a Flame Ionization Detector (FID) and an HP-5 capillary column identical to the previous one. The oven temperature profile remained consistent with the conditions above. The injector and detector temperatures were set at 230 °C and 240 °C, respectively. The quantification, presented as relative percentages, was carried out following the procedure established by *Qoreishi et al. (2023)*.

### Chlorophyll and carotenoid

During the flowering stage, fresh leaf samples weighing 0.5 g were subjected to homogenization in 10 ml of 80% acetone. The resultant homogenate underwent centrifugation at 12,000 g and a temperature of 4 °C for 15 min. Subsequently, the supernatant was measured at wavelengths of 663 nm, 645 nm, and 470 nm, employing a UV spectrophotometer (UV-1800, Shimadzu, Tokyo, Japan).

The quantification of chlorophyll *a*, chlorophyll *b*, and carotenoid content was accomplished using the equations established by *Lichtenthaler (1987)*. The units for plant pigments were expressed as milligrams per gram of fresh weight.

$$Ch1a = (12.25 A_{663.2}) - (2.79 A_{646.8}) \qquad (1)$$

$$Ch1b = (21.5 A_{646.8}) - (5.1 A_{663.2}) \qquad (2)$$

$$Car = \frac{[1000 A470 - 1.82 Ca - 85.02 Cb]}{198}. \qquad (3)$$

### *Relative water content (RWC)*

The weight of ten mature leaves was determined as their initial fresh weight (LFW). Subsequently, these samples were placed in distilled water at a temperature of 4 °C, and the weight of the turgid leaves was recorded after 24 h (LTW). Following this, the leaves were dried in an oven set at 70 °C for 48 h, and their weight as dry leaves was measured (LDW). Lastly, the relative water content (RWC) was computed using the formula provided by *Levitt (1980)*.

$$RWC(\%) = \frac{LFW - LDW}{LTW - LDW} \times 100. \tag{4}$$

## Proline

Fresh leaf samples weighing 0.5 grams each were pulverized using liquid nitrogen. These ground samples were combined with a 10 ml solution containing 3% sulfosalicylic acid. The mixture was centrifuged at 12,000 times the force of gravity (g) for 10 min. Following centrifugation, 2 ml of the resulting supernatant was carefully transferred into a fresh tube. To this supernatant, 2 ml of acid-ninhydrin solution and 2 ml of glacial acetic acid were added and thoroughly mixed.

The resulting mixtures were then subjected to heat within a water bath set at 100 °C for one hour. Subsequently, the mixtures were rapidly cooled by placing them in an ice bath. A total of 4 ml of toluene were introduced to the cooled mixtures, and vigorous vortexing was performed for 20 s. Finally, the supernatant was analyzed spectrophotometrically at a wavelength of 520 nanometers, with the proline content being quantified and expressed as micromoles per gram of fresh weight using the methodology established by *Bates, Waldren & Teare, 1973*.

## Total soluble sugars

The phenol and sulfuric acid methods were used to measure soluble sugar content. A total of 0.5 g of fresh leaves were homogenized with ethanol and mixed with 98% sulfuric acid and 5% phenol. This mixture was left for one hour, and its absorption was measured by a spectrophotometer at 485 nm (*Irigoyen, Einerich & Sánchez-Díaz, 1992*).

## Statistical analysis

Analysis of variance for the results was performed using the generalized linear model (GLM) model SAS 9.1.3 (SAS Institute Inc., Cary, NC, USA) software and the effects of irrigation regimes, the application of biofertilizers and the interactions between these two variables were evaluated by ANOVA. Means were compared using Duncan's multiple range test, and differences between individual means were considered significant at $p < 0.05$. The graphs were drawn in Excel.

## RESULTS

The experimental factors' interaction was significant regarding plant height, leaf count, chlorophyll $a$ and $b$ concentrations, total soluble solids (TSS), and proline content. Additionally, the main effect of irrigation and biofertilizer emerged as significant on

parameters such as dry yield, essential oil (EO) content, EO yield, carotenoid content, and relative water content (RWC).

## Morphological traits
### Plant height

The tallest plants, measuring 65 cm, were observed under optimal irrigation conditions (WW) when both AMF and PGPR were applied simultaneously. Nonetheless, no noteworthy contrast in plant height was observed with the discrete application of these two biofertilizers. Conversely, the most diminutive plants, standing at 37 cm, were observed in severe water stress (SWS), devoid of any biofertilizer application.

Relative to well-watered conditions (WW), the plant height experienced reductions of 11.8% and 26.9% under moderate water stress (MWS) and severe water stress (SWS), respectively. Furthermore, the introduction of AMF, PGPR, and the combination of AMF + PGPR yielded enhancements in plant height by 17.3%, 11.4%, and 28.9%, respectively, when contrasted against the control group (Fig. 2).

### Number of leaves

Simultaneous co-inoculation with both AMF and PGPR, alongside the well-watered (WW) treatment, exhibited a significant augmentation in the leaf count compared to the control group. The most minimal leaf count was registered within the SWS, wherein no biofertilizers were administered. Notably, leaf count exhibited 11.5% and 31.1% reductions under moderate water stress (MWS) and severe water stress (SWS), respectively. Conversely, the introduction of AMF, PGPR, or the combined application of AMF + PGPR led to increments of 17%, 8.5%, and 23.3%, respectively, in the number of leaves compared to the control group (Fig. 3).

### Dry yield

The highest dry yield (DY) of 11.53 grams per plant was observed under well-watered (WW) conditions, whereas the lowest DY of 8.71 grams per plant was documented in severe water stress (SWS) conditions. The DY exhibited 10.2% and 24.5% reductions under moderate water stress (MWS) and severe water stress (SWS), respectively.

The findings unveiled that the introduction of AMF, PGPR, or the combined application of AMF + PGPR could foster an elevation in DY by 23.7%, 13.5%, and 37.3%, respectively, compared to the control group (Table 1).

### Essential oil content

The maximum essential oil (EO) content, measuring 1.29%, was observed within the MWS treatment, while the lowest content of 1.01% was recorded in the WW treatment. Notably, the EO productivity of *Dorema kotschyi* Boiss. exhibited an increase of 27.7% and 13.9% in conditions characterized by MWS and SWS, respectively. Furthermore, the concurrent inoculation of AMF and PGPR markedly amplified the EO content of *D. kotschyi* Boiss. This enhancement was 27.2% compared to situations where biofertilizers were not applied (Table 2).

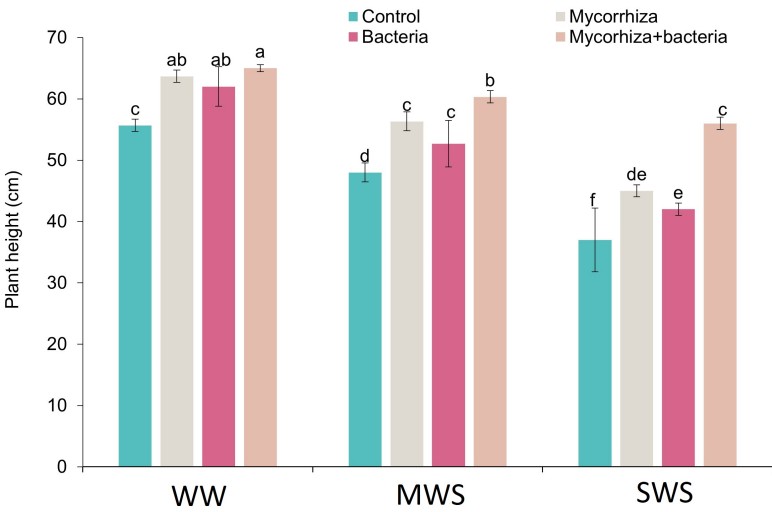

**Figure 2** **The plant height of *D. kotschyi* Boiss. under different soil moisture and fertilizers applications.** Using the Duncan test, similar letter(s) are not significantly different at a 5% probability level. WW, MWS and SWS correspond to well watering (100% field capacity), moderate water stress (80% field capacity), and severe water stress (60% field capacity), respectively.

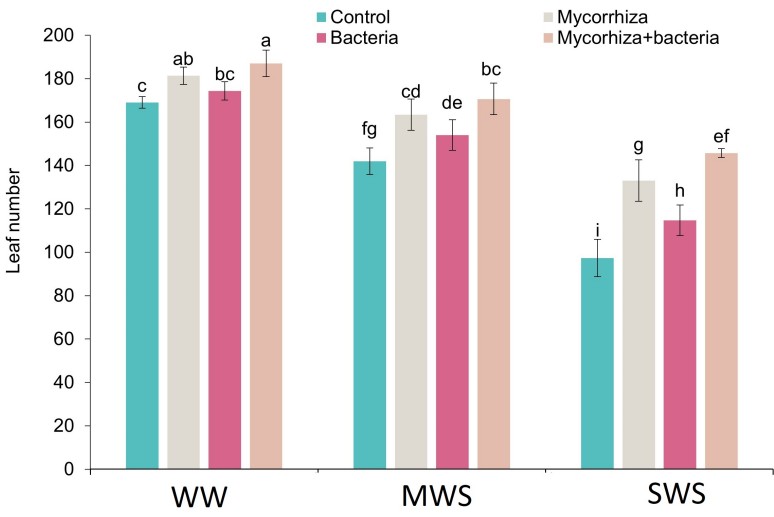

**Figure 3** **The number of leaves of *D. kotschyi* Boiss. under different soil moisture and fertilizers applications.** Using the Duncan test, similar letter(s) are not significantly different at a 5% probability level. WW, MWS and SWS correspond to well watering (100% field capacity), moderate water stress (80% field capacity), and severe water stress (60% field capacity), respectively.

## Essential oil yield

The highest essential oil (EO) yield, measuring 0.13 grams per plant, was achieved under conditions of MWS, while the lowest yield of 0.10 grams per plant was observed in SWS treatments. Additionally, the most substantial EO yield was documented when the

**Table 2** The main effects of different soil moistures and fertilizers on the dry yield, essence content and essence yield of *Dracocephalum kotschyi* L.

| | | Dry yield (g plant$^{-1}$) | Essence content (%) | Essence yield (g plant$^{-1}$) |
|---|---|---|---|---|
| Soil moisture | | | | |
| | WW | 11.53 ± 1.233[a] | 1.01 ± 0.17[b] | 0.12 ± 0.03[b] |
| | MWS | 10.36 ± 1.39[b] | 1.29 ± 0.29[a] | 0.13 ± 0.04[a] |
| | SWS | 8.71 ± 1.66[c] | 1.15 ± 0.29[ab] | 0.10 ± 0.04[c] |
| Fertilizers | | | | |
| | Control | 8.6 ± 1.19[d] | 1.03 ± 0.19[c] | 0.09 ± 0.03[c] |
| | AMF | 10.64 ± 1.70[b] | 1.14 ± 0.10[b] | 0.12 ± 0.02[b] |
| | PGPR | 9.76 ± 1.15[c] | 1.11 ± 0.18[b] | 0.11 ± 0.02[bc] |
| | AMF + PGPR | 11.81 ± 1.65[a] | 1.31 ± 0.33[a] | 0.15 ± 0.03[a] |

Notes.

The results are expressed as means of three replicates ± standard error (SE).

WW, MWS and SWS correspond to well watering (100% field capacity), moderate water stress (80% field capacity), and severe water stress (60% field capacity), respectively.

Means in each column followed by a similar letter(s) are not significantly different at a 5% probability level using the Duncan test.

co-inoculation of AMF and PGPR was implemented, resulting in a noteworthy increase of 66.7% compared to the control treatment (Table 2).

## Chlorophyll content

The highest concentrations of chlorophyll *a* (0.59 mg g$^{-1}$ fresh weight) and chlorophyll *b* (0.24 mg g$^{-1}$ fresh weight) were observed in plants subjected to WW conditions and co-inoculated with both AMF and PGPR. Compared to the well-watered conditions, the chlorophyll *a* and *b* contents experienced 12.7% and 13.6% under MWS and reductions of 21.8% and 22.7% under SWS treatments. Remarkably, the co-inoculation of AMF and PGPR yielded a substantial enhancement in chlorophyll *a* and *b* content, increasing by 32.5% and 31.2%, respectively (Figs. 4A and 4B).

## Carotenoids

The most elevated carotenoid concentration (2.22 mg g$^{-1}$ fresh weight) was derived from plants subjected to MWS, closely trailed by those under SWS, with concentrations 15.6% and 9.4% higher than WW conditions, respectively. Moreover, the introduction of AMF, PGPR, and their combined application (AMF + PGPR) all contributed to enhancements in carotenoid concentration, specifically by 4.6%, 2.5%, and 13.6%, respectively (Table 3).

## Relative water content

The maximum relative water content (RWC), measuring 88.73%, was observed in WW conditions, while the lowest content of 72.48% was recorded under SWS. The RWC exhibited a downward trend in conjunction with diminishing soil water availability. Compared to WW treatment, the RWC content saw 10.7% and 18.3% reductions in the presence of MWS and SWS, respectively. Remarkably, the co-inoculation of AMF and PGPR resulted in the highest RWC value, surpassing the control treatment by 15.5% (Table 3).

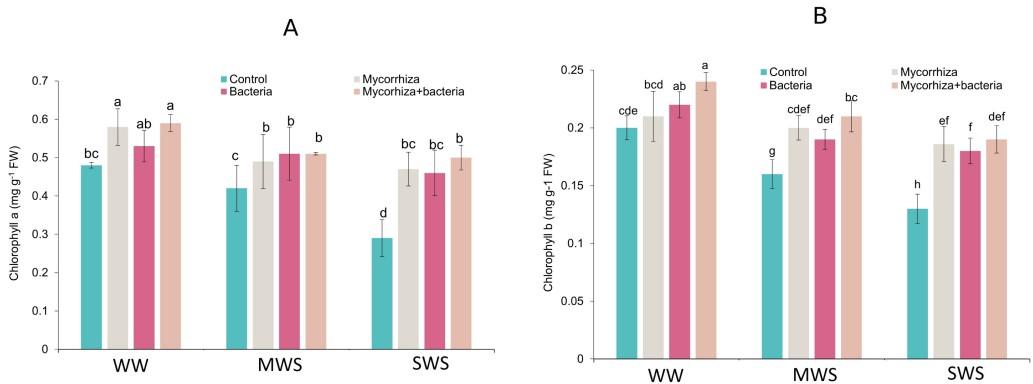

**Figure 4 The chlorophyll a and b of *D. kotschyi* Boiss. under different soil moisture and fertilizers applications.** Using the Duncan test, similar letter(s) are not significantly different at a 5% probability level. WW, MWS and SWS correspond to well watering (100% field capacity), moderate water stress (80% field capacity), and severe water stress (60% field capacity), respectively.

**Table 3 The main effects of different soil moistures and fertilizers on the carotenoid and relative water content (RWC) of *Dracocephalum kotschyi* L.**

| | | Carotenoids (Mg g$^{-1}$ fresh weight) | RWC (%) |
|---|---|---|---|
| Soil moisture | | | |
| | WW | 1.92 ± 0.09[c] | 88.73 ± 4.94[a] |
| | MWS | 2.22 ± 0.18[a] | 79.21 ± 4.36[b] |
| | SWS | 2.10 ± 0.134[b] | 72.48 ± 5.67[c] |
| Fertilizers | | | |
| | Control | 1.98 ± 0.205[c] | 79.12 ± 6.77[c] |
| | AMF | 2.07 ± 0.128[b] | 87.88 ± 7.61[b] |
| | PGPR | 2.03 ± 0.144[bc] | 81.15 ± 8.65[c] |
| | AMF + PGPR | 2.25 ± 0.162[a] | 91.38 ± 8.20[a] |

**Notes.**

The results are expressed as means of three replicates ± standard error (SE), WW, MWS and SWS corresponding to well watering (100% field capacity), moderate water stress (80% field capacity), and severe water stress (60% field capacity), respectively.

Means in each column followed by a similar letter (s) are not significantly different at a 5% probability level using the Duncan test.

## Total soluble sugars

The simultaneous application of AMF and PGPR and treatment of SWS exhibited a marked increase in total soluble solids (TSS) content. Notably, the TSS content demonstrated an upward trajectory as soil water availability diminished. Compared to treatment WW, the TSS content experienced 32.8% and 68.5% increments under MWS and SWS conditions, respectively. Intriguingly, the inoculation with AMF, PGPR, and the combined application of AMF + PGPR led to enhancements in TSS content by 20.2%, 17.2%, and 38.1%, respectively (Fig. 5).

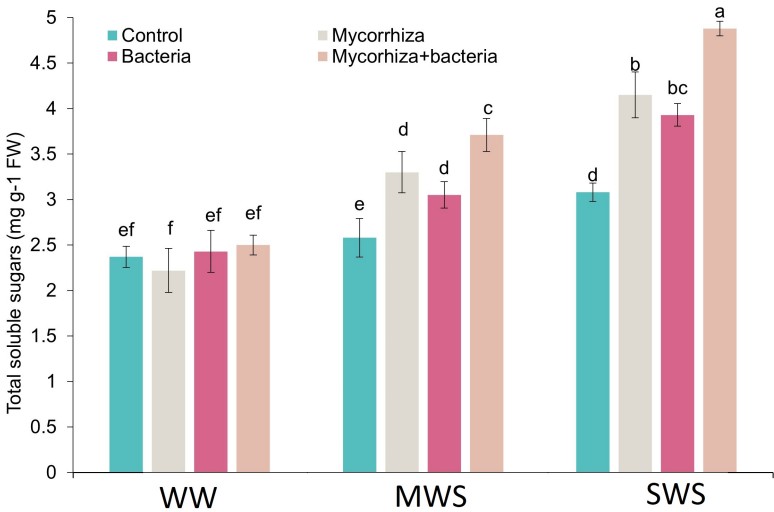

**Figure 5  The total soluble sugars of *D. kotschyi* Boiss. under different soil moisture and fertilizers applications.** Using the Duncan test, similar letter(s) are not significantly different at a 5% probability level. WW, MWS and SWS correspond to well watering (100% field capacity), moderate water stress (80% field capacity), and severe water stress (60% field capacity), respectively.

## Proline

The simultaneous application of AMF and PGPR, coupled with the SWS treatment, yielded the highest proline content (18.5 μmol g$^{-1}$ fresh weight). The proline content exhibited an ascending trend as soil water availability decreased. Compared to WW conditions, the proline content demonstrated 16.6% and 33.1% increments under MWS and SWS treatments, respectively.

Furthermore, the introduction of AMF, PGPR, and the combined application of AMF + PGPR led to enhancements in proline content by 13.1%, 4.7%, and 26.5%, respectively (Fig. 6).

## Phytochemical profile

Applying GC-MS and GC-FID analysis, 26 compounds were successfully identified within the essential oil (EO) of *Dorema kotschyi* Boiss. Among these, the predominant constituents were geranyl acetate (11.4–18.88%), alpha-pinene (9.33–15.08%), Bis(2-Ethylhexyl) phthalate (8.43–12.8%), neral (6.80–9.32%), geranial (9.23–11.91%), and limonene (5.56–9.12%). Remarkably, the highest levels of geranyl acetate, geranial, limonene, and alpha-pinene were observed within the context of MWS, particularly following the co-application of AMF and PGPR. Similarly, the maximum content of Bis(2-Ethylhexyl) phthalate and neral was documented under SWS, specifically when the co-inoculation of AMF and PGPR was implemented (Table 4).

## DISCUSSION

The study's findings revealed a noticeable decline in yield and associated components because of the water deficit treatments: moderate water stress (MWS) and severe water

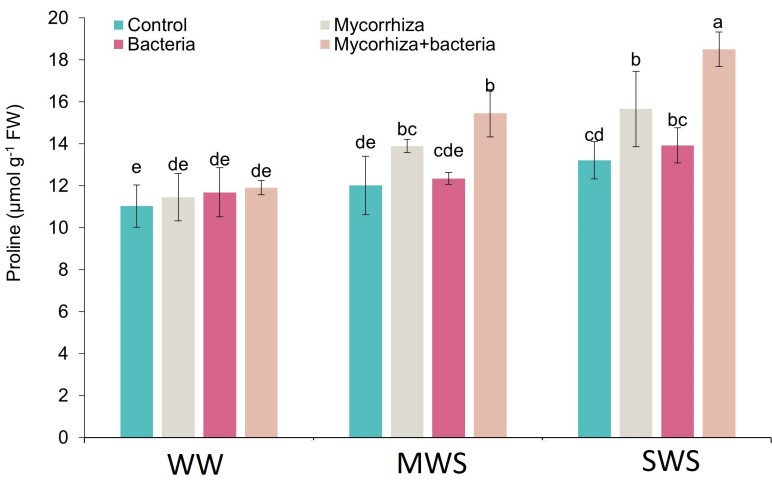

**Figure 6** **The proline content of *D. kotschyi* Boiss. under different soil moisture and fertilizers applications.** Using the Duncan test, similar letter(s) are not significantly different at a 5% probability level. WW, MWS and SWS correspond to well watering (100% field capacity), moderate water stress (80% field capacity), and severe water stress (60% field capacity), respectively.

stress (SWS). In situations of moderate water limitation, the reduction in carbon dioxide ($CO_2$) absorption and photosynthesis rate can be attributed to the stomatal closure mechanism, which serves to curtail water transpiration and, subsequently, the production of plant biomass (*Mathobo, Marais & Steyn, 2017*). Under conditions of severe water scarcity, the decrement in plant yield can be attributed to disruptions in chloroplast structure coupled with diminished cell division and elongation. This cascade of effects leads to compromised leaf development and expansion, diminishing light interception and photosynthesis rate (*Zhang et al., 2022*).

Furthermore, the study's results align with observations made by *Cham et al. (2022)*, who noted that applying biofertilizers enhanced the growth and dry weight of *Dorema kotschyi* Boiss. Incorporating biofertilizers was also found to bolster the plant's resilience against water scarcity, diseases, and pests.

The co-application of AMF and PGPR emerges as the most efficacious treatment, yielding the highest agronomic traits and dry yield. The introduction of AMF contributes to enhanced nutrient and water absorption, thereby improving primary plant metabolism and elevating both photosynthesis rate and overall performance (*Golubkina et al., 2020*; *Heydarzadeh et al., 2022*). Similarly, the application of PGPR induces a stimulatory effect on plant growth, primarily by augmenting the rate of nitrogen fixation, enhancing nutrient solubility—particularly for phosphorus (P) and potassium(K)—promoting siderophore production, and facilitating the synthesis of enzymes that regulate ethylene production (*Faridvand et al., 2022*). In another research, it was found that the presence of suitable conditions for the growth of *D. moldavica* plants, including increasing the availability of necessary nutrients, especially nitrogen, under the influence of co-application AMF with PGPR under water deficit stress, through improving growth, development, and photosynthesis, leads to an improvement in plant height, and leaves number, which

ultimately leads to an increase in the yield of dry weight (*Ghanbarzadeh et al., 2019*). Inoculated plants also improved stomatal regulation through hormonal adjustment (*Pagnani et al., 2018*). Production of IAA by symbiotic bacteria may partly explain their effectiveness in promoting plant growth under adverse environmental conditions (*Pagnani et al., 2018*). It has been reported that the inoculation of biofertilizers enhances growth parameters and aerial dry weight in *D. kotschyi* Boiss. under water deficit stress (*Shaabani et al., 2020*). *Heydarzadeh et al. (2023b)* reported that applying biofertilizer improved the biological and grain yields of dragon head plants compared to those not subjected to fertilization under water stress. *Zamani et al. (2023)* reported that AMF and PGPR applications improved plant development and production of dragon head plants under water shortage stress states by increasing nutrient accessibility and absorption. The authors attributed such results to the positive effect exerted by the treatments on vegetative growth, chlorophyll synthesis, and photosynthesis capacity, especially under water shortage tension (*Ghanbarzadeh et al., 2019*). The beneficial influence of AMF and PGPR may be related to increased soil organic matter content and a regulated availability of nutrients in the agricultural soil, which directly influence dragon head plant photosynthetic and vegetative growth (*Heydari & Pirzad, 2020*).

The application of both AMF and PGPR, denoted as treatment MWS, led to an enhancement in essential oil (EO) biosynthesis, EO yield, and the major constituents of the EO, namely geranyl acetate, geranial, limonene, and alpha-pinene. This augmentation in EO production under stressful conditions underscores the plant's adaptive physiological responses to challenging environments.

In situations characterized by water limitation, the photosynthesis rate tends to decrease due to the closure of stomata, which in turn reduces the uptake of carbon dioxide ($CO_2$). This leads to the accumulation of NADPH and H+ ions within cells. In response, the biosynthesis of EO compounds is activated as an alternative pathway, specifically through methyleritrophosphate and mevalonic acid pathways, which consume excess NADPH and H+ ions (*Kaur & Suseela, 2020*; *Wang, Quan & Xiao, 2019*). Similar phenomena have been documented in other plant species; for example, the essential oil yield and major EO compounds of *Rosmarinus officinalis* L. were reported to be increased under moderate drought stress conditions (*Abbaszadeh et al., 2020*). The elevation in the biosynthesis of EO constituents resulting from the combined application of AMF and PGPR can be attributed to their role in enhancing nutrient accessibility. This enhancement subsequently facilitates the development of glandular trichomes and the formation of EO intermediate compounds such as ATP and acetyl coenzyme A (*Alipour et al., 2021*; *Eshaghi Gorgi et al., 2022*). *Heydarzadeh et al. (2023b)* reported that applying biological fertilizers improved the oil content and oil yield of the dragon head plants under water shortage stress. Also, such treatments may enhance the extent of essential oil-producing glands in flowers and leaves of the dragon head plant under water shortage stress. In the current investigation, biofertilizers such as AMF and PGPR may have enhanced the nutrient uptake and improved plant moisture relationships compared to control plants, leading to the essential oil and essential oil yield of the dragon head plants increment (*Heydari & Pirzad, 2020*).

Peerj

**Table 4** Effect of irrigation regime and biofertilizer on essential oil compounds of *Dracocephalum kotschyi* Boiss.

| No | Component | RI | Treatments | | | | | | | | | | | |
|---|---|---|---|---|---|---|---|---|---|---|---|---|---|---|
| | | | WW C | WW AMF | WW PGPR | WW AMF + PGPR | MWS C | MWS AMF | MWS PGPR | MWS AMF + PGPR | SWS C | SWS AMF | SWSP GPR | SWS AMF +PGPR |
| 1 | Alpha-pinene | 928 | 9.33 | 10.64 | 9.77 | 10.99 | 12.83 | 13.07 | 14.88 | 15.08 | 12.22 | 13.69 | 14.03 | 14.64 |
| 2 | Sabinene | 965 | 2.62 | 1.62 | 0.7 | 0.22 | 1.12 | 2.11 | 0.09 | 0.01 | 2.1 | 1.12 | 0.09 | 0.02 |
| 3 | Beta-pinene | 969 | 1.39 | 1.32 | 0.01 | 0.38 | – | 1.08 | – | 0.02 | 2.00 | 0.01 | 0.01 | – |
| 4 | Beta-myrcene | 981 | 2.5 | 1.1 | 0.06 | 0.38 | 0.04 | 1.02 | 0.36 | 0.01 | 2.09 | 0.11 | 1.5 | – |
| 5 | P-cymene | 1018 | 1.7 | 1.51 | 0.8 | 0.67 | 0.83 | 1.61 | 0.74 | 0.64 | – | 1.00 | 1.7 | 0.51 |
| 6 | Limonene | 10.25 | 5.56 | 6.18 | 5.44 | 6.11 | 7.66 | 8.32 | 8.3 | 9.12 | 5.9 | 6.19 | 7.56 | 7.98 |
| 7 | 1,8-Cineole | 1026 | – | 0.01 | 0.7 | 0.49 | 0.82 | – | 0.61 | 0.05 | – | 0.01 | 0.01 | 0.04 |
| 8 | Cis-sabinene hydrate | 1060 | 0.88 | – | 0.58 | 0.37 | 0.86 | 1.10 | 0.03 | 0.01 | 2.01 | 0.03 | 0.88 | – |
| 9 | Trans-Sabinene hydrate | 1082 | 1.72 | 1.02 | 1.03 | 0.88 | 0.76 | 1.46 | 0.93 | 0.85 | 1.1 | 2.18 | 0.72 | 1.02 |
| 10 | P-Linalool | 1092 | 2.39 | 1.45 | 1.44 | 1.37 | 2.31 | 2.7 | 1.44 | 0.92 | 3.38 | 1.18 | 1.39 | 1.45 |
| 11 | Alpha-Campholenal | 1120 | 3.32 | 4.37 | 4.85 | 4.21 | 2.93 | 2.43 | 2.06 | 0.66 | 2.57 | 2.09 | 2.32 | 2.37 |
| 12 | Trans-p-mentha-2,8-dien-1 | 1129 | 1.37 | – | – | 0.04 | 0.69 | 1.01 | 0.10 | – | 2.00 | 0.06 | 0.37 | – |
| 13 | Trans-Pinocarveol | 1134 | 1.08 | 1.91 | 1.98 | 1.61 | 1.34 | 1.52 | 1.75 | 0.53 | 1.22 | 1.00 | 1.08 | 1.91 |
| 14 | Trans-Verbenol | 1140 | 2.31 | 3.06 | 3.19 | 2.66 | 1.66 | 1.57 | 1.03 | 0.83 | 1.92 | 2.41 | 2.31 | 3.06 |
| 15 | Pinocarvone | 1157 | 0.6 | 1.7 | 0.74 | 0.63 | 0.53 | | 0.7 | 0.53 | 1.00 | 0.48 | 1.6 | 0.7 |
| 16 | Terpinene-4-ol | 1171 | 1.66 | 0.55 | 0.85 | 0.48 | 1.15 | 0.46 | 0.44 | 0.43 | 1.16 | 2.83 | 1.66 | 0.55 |
| 17 | Alpha-terpineol | 1185 | 0.61 | 1.7 | 0.66 | 0.5 | – | – | 0.54 | 0.04 | 1.00 | 2.17 | 1.61 | 0.7 |
| 18 | Trans-Carveol | 1214 | 1.76 | 1.99 | 2.25 | 1.88 | 0.8 | 0.84 | 1.1 | 0.98 | 0.29 | 1.53 | 1.76 | 1.99 |
| 19 | Neral | 1238 | 8.42 | 8.32 | 9.19 | 8.93 | 8.89 | 8.5 | 8.17 | 8.97 | 6.8 | 6.94 | 7.42 | 9.32 |
| 20 | Geraniol | 1250 | 4.44 | 3.16 | 3.87 | 3.93 | 3.45 | 4.88 | 4.48 | 2.44 | 3.3 | 4.08 | 3.44 | 5.16 |
| 21 | Geranial | 1269 | 9.83 | 10.68 | 10.2 | 10.93 | 9.99 | 10.36 | 11.15 | 11.91 | 9.57 | 9.23 | 10.83 | 10.68 |
| 22 | Bornyl acetate | 1280 | 0.44 | 1.63 | 0.5 | 0.45 | 0.44 | – | 0.53 | 0.49 | 1.00 | 2.00 | 0.44 | 0.63 |
| 23 | Methyl geranoate | 1317 | 6.29 | 6.21 | 6.99 | 6.41 | 6.7 | 6.47 | 6.46 | 7.58 | 4.83 | 5.11 | 6.29 | 6.91 |
| 24 | Geranyl acetate | 1378 | 11.4 | 13.45 | 13.36 | 13.83 | 13.66 | 15.48 | 16.48 | 18.88 | 12.4 | 11.9 | 13.4 | 15.45 |
| 25 | Phytol | 2093 | 1.55 | 1 | 0.4 | – | 1.06 | 0.56 | 0.37 | 0.67 | 3.69 | 1.4 | 2.55 | 0.1 |
| 26 | Bis (2-ethylhexyl) phthalate | 2537 | 8.43 | 10.8 | 11.77 | 11.99 | 9.17 | 10.72 | 10.92 | 12.69 | 10.56 | 13.31 | 8.43 | 12.8 |
| | Total | | 91.6 | 95.38 | 91.33 | 90.34 | 89.69 | 97.27 | 93.66 | 94.34 | 94.11 | 92.06 | 93.4 | 97.99 |

**Notes.**

[a]RI, linear retention indices on DB-5 MS column, experimentally determined using homolog series of n-alkanes, data are mean ± SE ($n = 3$); the main components are shown in bold.

WW, MWS and SWS correspond to well watering (100% field capacity), moderate water stress (80% field capacity), and severe water stress (60% field capacity), respectively.

Chlorophyll content exhibited a pronounced decline in response to increased water availability, primarily attributable to the occurrence of membrane lipid peroxidation and the decomposition of chlorophyll due to reactive oxygen species (ROS) (*Rezaei-Chiyaneh et al., 2021a*). Observations in lettuce (*Lactuca sativa* L.) seedlings have similarly indicated reduced chlorophyll *a* and *b* content under drought stress conditions (*Shin et al., 2021*). Conversely, the carotenoid content demonstrated enhancement following both MWS and SWS treatments. This phenomenon can be attributed to the operation of both enzymatic and non-enzymatic antioxidant systems, which effectively counteract oxidative damage in plant tissues subjected to water limitation. Among the non-enzymatic antioxidants, carotenoids (including xanthophylls and carotenes), along with ascorbates and alpha-tocopherol, play a vital role in bolstering plant tolerance against stress by attenuating the activity of ROS compounds (*Farooq et al., 2019*). The increased carotenoid concentration observed in response to water deficit stress conditions reinforces plant defence mechanisms to cope with stress-induced challenges. *Paravar, Farahani & Rezazadeh (2021)* reported that by enhancing the water shortage stress in the dragon head, chlorophyll a, b, chlorophyll a + b, and carotenoids at flowering stages were decreased; nevertheless, they were improved in response to AMF and PGPR. The degradation of such pigments or the reduction of their synthesis, associated with the reduced activity of enzymes involved in their synthesis, causes chlorophyll decline in dragon head plants exposed to water deficit stress, resulting in reduced assimilation material and, thus, performance losses (*Heydarzadeh et al., 2023b*).

The relative water content (RWC) exhibited a reduction concomitant with the diminishing availability of soil water. This decline in RWC in response to water deficit stress conditions arises from an escalation in leaf transpiration rate coupled with a decrease in the rate of water absorption by the roots. Notably, the integrated application of AMF and PGPR contributed to an enhancement in RWC. The mechanism behind this enhancement is closely linked to the penetration of fungal hyphae into the root cortex and endoderm, which augments the hydraulic conductivity of water by establishing a path characterized by low resistance across the root cells. This phenomenon has been noted in prior research, where the inoculation of *Dorema moldavica* L. roots with AMF enhanced RWC under water deficit stress conditions (*Ghanbarzadeh et al., 2019*). *Paravar, Maleki Farahani & Rezazadeh (2022)* reported that the leaf RWC of the dragon head diminishes as water stress increases. The decrease of tissues' turgor in plant and leaf RWC can be the first effect of water deficit stress, which can have a natural influence on the development, growth, and ultimate size of cells (*Faridvand et al., 2022*).

AMF and PGPR fertilizers enhance water uptake in the host plant by altering root architecture and spreading the plant's root system (*Zamani et al., 2023b*). Indeed, biofertilizers can alleviate the negative influences of water deficit stress on dragon head plants by enhancing leaf moisture potential, transpiration rate, photosynthetic efficiency, and rate of $CO_2$ use (*Heydari & Pirzad, 2020*). Moreover, they can promote the absorption of nutrients, enhancing growth and plant production. In the present study, an augmentation in proline and total soluble solids(TSS) content was observed under drought-stress conditions following the application of AMF and PGPR. These concentrations of TSS substances allow growth in low water conditions. The most
important source of TSS substances is photosynthetic compounds, which are either produced directly or from the hydrolysis of carbon reserves (*Heydarzadeh et al., 2023a*). Since both photosynthesis and growth are affected by drought stress, their balance affects the accumulation of TSS substances (*Paravar, Maleki Farahani & Rezazadeh, 2022*). Considering that a water shortage affects growth before photosynthesis is affected, the accumulation of photosynthesis products seems obvious (*Ghanbarzadeh et al., 2019*). Therefore sucrose acts as a signaling molecule in low concentrations and as a scavenger of reactive oxygen species in high concentrations (*Javanmard et al., 2022*). Thus the buildup of TSS inside the cells performs a necessary part in the regulation of osmosis and permits the reduction of the water potential of the cell, so more water remains inside the cell to maintain turgor under drought stress (*Rezaei-Chiyaneh et al., 2021a*).

In drought stress conditions, osmotic regulators can escalate the power of water uptake by plant cells. The accumulation of this osmotic regulator can be related to drought resistance (*Shaabani et al., 2020*). The improved vegetative growth could account for the higher TSS in applying AMF and PGPR-treated plants of dragon heads (*Heydari & Pirzad, 2020*). The application of AMF and PGPR probably enhanced growth and led to a greater concentration of carbohydrates under drought stress. This elevation in the levels of osmolytes like proline and TSS is recognized as a vital adaptive response of plants to counteract the impact of drought and other stressful circumstances. The accumulation of these compounds during stressful conditions plays a crucial role in preserving leaf cell turgidity, safeguarding cell membranes, preventing protein denaturation, and neutralizing free radicals (*Furlan et al., 2020*). The increase in TSS in the application of AMF and PGPR, which affects the osmotic potential, helps to maintain the health and function of the cell membranes of the dragon head that have been damaged by water stress (*Zamani et al., 2023b*).

Reinforcing this observation, it has been documented that the proline content in *Lallemantia iberica* plants increased notably under moderate and severe drought stress (*Javanmard et al., 2022*). Additionally, the increased concentration of osmolytes within plant cells under water-limiting conditions serves to lower the osmotic potential, thereby enhancing the plant's capacity to absorb water from the soil *via* the roots, consequently boosting its resilience in such situations (*Rezaei-Chiyaneh et al., 2023*). Moreover, the enhanced proline content resulting from the integrated application of AMF + PGPR can be attributed to the intertwined relationship between proline concentration in plant cells and nitrogen availability. The accessibility of nutrients, especially nitrogen, is facilitated by the role of these fertilizers. Additionally, it has been reported that the symbiotic association between AMF and plant roots under drought stress conditions promotes an increase in proline content through the upregulation of the glutamate pathway and downregulation of proline degradation (*Thangaraj et al., 2022*). This interplay between plant and AMF has been evident in sweet potatoes (*Ipomoea batatas* L.), where AMF inoculation led to increased tolerance to water deficit by elevating proline content and soluble sugars (*Yooyongwech et al., 2016*). Dragon head plants inoculated with biofertilizers such as AMF + PGPR can usually manage nutrients and water more suitably than non-treated crops under moisture shortage stress situations (*Paravar, Maleki Farahani & Rezazadeh, 2022*).

Therefore, the proline content, such as dragon head plants, indicates a lower upsurge compared to non-treated plants.

## CONCLUSIONS

The study reveals the detrimental impact of water stress on the agronomic traits and dry matter yield of *Dorema kotschyi* Boiss. plant. However, in the face of water stress conditions, the concurrent application of AMF and PGPR emerges as a strategy that effectively mitigates these adverse effects. This integrated approach showcases its potential to enhance various parameters, including chlorophylls, carotenoids, proline, and total soluble sugars. Furthermore, the co-application of AMF + PGPR positively influences the dry matter yield and essential oil yield of *D. kotschyi* Boiss. plants, marking a significant achievement in pursuit of the primary cultivation goal.

The synergistic effect of AMF and PGPR is observed in the elevation of essential oil content and the pivotal compounds within the essential oil, notably geranyl acetate, geranial, limonene, and alpha-pinene. This promising outcome holds substantial implications for enhancing grain and essential oil yields while concurrently addressing environmental concerns linked to the excessive usage of chemical fertilizers. Moreover, the cultivation of *D. kotschyi* Boiss., known for its low water requirements, presents an avenue for prudent water management and increased plant tolerance. In light of these findings, the co-application of AMF and PGPR in water stress conditions emerges as a viable strategy to achieve the dual objectives of augmenting plant productivity and essential oil yield while concurrently advancing sustainable agriculture principles.

In contrast to earlier investigations into *Dracocephalum kotschyi*, this study has made a substantial contribution to our comprehension of how the combined application of AMF and PGPR can improve the plant's growth and biochemical characteristics when subjected to water scarcity. Notably, this study has shed light on the mechanisms behind the observed improvements. The choice of AMF and PGPR species does matter, as the co-application of certain strains yielded the most promising results. However, there is a need for further exploration and identification of the most effective strains for *Dracocephalum kotschyi*.

The next step in this research area should involve a more comprehensive investigation into the specific AMF and PGPR strains that can maximize the plant's resilience to water scarcity. Additionally, understanding the molecular and genetic mechanisms underlying the observed improvements in growth and biochemical attributes would be valuable. This knowledge gap presents an exciting opportunity for future research, which could lead to the development of tailored strategies for enhancing the growth of *Dracocephalum kotschyi* under water deficit conditions. Furthermore, while this study has shown improvements in yield, essential oil production, and physiological responses, more extensive field trials and long-term assessments are necessary to confirm the practical applicability of these findings in real-world agricultural settings.

### Funding

The authors received no funding for this work.

### Competing Interests

The authors affirm that they have no conflicts of interest to declare.

### Author Contributions

- Saeid Gasemi conceived and designed the experiments, performed the experiments, analyzed the data, prepared figures and/or tables, authored or reviewed drafts of the article, and approved the final draft.
- Hassan Mahdavikia conceived and designed the experiments, performed the experiments, analyzed the data, authored or reviewed drafts of the article, and approved the final draft.
- Esmaeil Rezaei-Chiyaneh conceived and designed the experiments, performed the experiments, analyzed the data, authored or reviewed drafts of the article, and approved the final draft.
- Farzad Banaei-Asl analyzed the data, prepared figures and/or tables, and approved the final draft.
- Aria Dolatabadian analyzed the data, prepared figures and/or tables, authored or reviewed drafts of the article, and approved the final draft.
- Amir Sadeghpour analyzed the data, authored or reviewed drafts of the article, and approved the final draft.

### Data Availability

The raw data are available in the Supplemental Files.

### Supplemental Information

Supplemental information for this article can be found online at http://dx.doi.org/10.7717/peerj.16474#supplemental-information.

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
