# Peer review of "Co-inoculation of mycorrhizal fungi and plant growth-promoting rhizobacteria improve growth, biochemical and physiological attributes in Dracocephalum kotschyi Boiss. under water deficit stress"

_PeerJ, doi:10.7717/peerj.16474_

## Round 0.1 · original submission · Major Revisions

Dear Dr. Mahdavikia and Dr. Dolatabadian,

Thank you for your submission to PeerJ.

Based on review reports and my own assessment, you have to thoroughly revise your manuscript to make it suitable for publication. The reviewers have suggested a suite of major changes to improve the quality of your paper. You are expected to carefully consider all these comments, and make the necessary changes. They have pointed out shortcomings in almost each and every section of the manuscript. As the English language is not up to the mark, you are advised to take the help of colleagues or professional editors in this regard.

Look forward to receiving the revised manuscript, with all the suggested changes made. The revised paper will again be evaluated to ensure that there are no mistakes.

·

Basic reporting

At first, I would to mention the actual topic of the study – to compare the impact of AMF and PGPR to the drought resistance of D. kotschyi.
Introduction section is very brief and doesn’t cover some previous studies on the topic (D. kotschyi drought tolerance):
https://doi.org/10.30495/ejmp.2021.694476
https://doi.org/10.30495/wsrcj.2021.18543
https://doi.org/10.3389/fpls.2023.1063618
https://doi.org/10.1007/s00344-020-10157-6
https://doi.org/10.22092/jmpb.2020.122075
https://doi.org/10.22059/ijhst.2019.280572.294
https://doi.org/10.1016/j.sajb.2022.04.008

Experimental design

I suggest to rename watering regimes I1 to WW (well watering), I2 to MWS (mild water stress), I3 to SWS (severe water stress) to better understanding.
Is it known what particular bacterial and fungal strains were used in biofertilizer (L134, L140)? The genomic features may differ among strains of the same specie.
What watering regime was applied for cases in Table 3 (lines 4-7)?
L200: Please describe details of statistical analysis, not only the software used.

Validity of the findings

The Discussion section contains no comparison with the previous studies. Most of this section devoted to the studies barely related to the object (D. kotschyi).
L289-291: comparison with different plant specie is questionable. Please choose the appropriate study no compare (see the list above).
L292-L301: this part better fits to the Introduction.
L315: comparison with the study (Amani Machiani et al., 2022) is incorrect – the different plant specie, different AMF, intercropping with soybean – experiment conditions differ significantly. The same problem for L331-L332: Agüero-Fernández et al. describes different plant specie and salt stress.
L348: cited study (Alotaibi et al., 2021) describes the aluminum toxicity case, not related with the watering regime.
My notes about the main text:
• L136: “fixing” → “fixation”

·

Basic reporting

Comment for the author:
General comment:
 The manuscript was poorly laid out and should be rejected at this time. There are numerous misunderstandings and typing errors. The manuscript's English must be greatly improved by senior authors, colleagues, or any fluent English speaker or by software like Grammarly. MS should be revised thoroughly before being submitted again for fresh review.
 Many words in whole MS especially genera and species name are clump together. Pls check.
Specific comment
 Title should be rewrite. Exclude the abbreviation from the title.
 Abstract is poorly laid out, not actually represented the actual significance of the works. Abstract should reflect the quantitative value of result. Therefore, it needs to be rewrite carefully including result of all the observed attributes.
 Include the company names and the nation from where the equipments/chemicals were purchased. Whenever applicable.
 Mentioned controlled experimental condition viz., light and dark hour per day, temperature, amount of water regularly given, humidity, longitude and latitude positions of places where the experiments were performed.
 How long did the experiment last? How to maintain the experimental condition including light, temperature, moisture etc? Mention duration of experiment? It is not clear.
 How many plants were grown in each pot? It should be mentioned in MS.
 What mean oven under shade? The temperature at which the sample kept in MS should be included in MS.
 The genera and species name in somewhere in MS not in italic. Pls check
 Method should be comprehensive and correct in order to repeat the work by other researchers.
 Mention the expression unit of many biochemical activities, whenever applicable.
 Mention about the standards and blank in biochemical activities, whenever applicable.
 In figure (chlorophyll a), the letter above the error bar not seems to show the actual result. Please recheck it or otherwise submit the raw data as supplementary file for understanding and cross validation.
 Kindly include the value of standard error of each treatment in table.
 Mentioned software name which was used for preparation of figure.
 Discussion is OK.
 CONCLUSION: Although conclusion is based on result obtained. In my opinion, it should be rewrite and reflect the novelty and future prospect more precisely
 References somewhere are not correct? Moreover, the genera name in reference should also be in italic.


Specific line comment
 L148: M&M section. Replace measurement heading with observation
 L166: Replace rpm with corresponding g value.
 L166: mentioned centrifuge temperature.
 L167: mentioned OD taken what? Supernatant or suspension.
 L168: Mentioned expression unit of plant pigments
 L178: Replace measured word with observed word
 L179: Replace 0C with ℃.
 L186: Replace rpm with corresponding g value.
 L190: Mentioned OD take after colour development.
 L200: Mentioned software to analyse data and calculate standard error.
 L211: Replace highest with tallest
 L227 & L232: Replace highest with maximum
 L227 & L232: Replace lowest with minimum

Experimental design

-

Validity of the findings

-

Additional comments

-

---

## Round 0.2 · Minor Revisions

Dear Dr. Mahdavikia and Dr. Dolatabadian,

Thank you for your submission to PeerJ.

Based on the reevaluation of your manuscript, I am of the view that it still needs a number of Minor Revisions. Accordingly, you have to get it revised as per the reviewer comments shown below.


With regards

·

Basic reporting

pass

Experimental design

pass

Validity of the findings

Discussion section still needs improvement. While there are many studies devoted to the Dracocephalum kotschyi species, authors might highlight what new knowledge was obtained in comparison with the previous studies. Is the particular species of AMF/PGPR does matter? What the next step in the topic, what knowledge gap remains in the field?

Additional comments

Authors simply answers "Done" or "Revised" for most of the comments, that incorrect response. Ypu should comment in details all changes in the manuscript (besides minor corrections).

The whole manuscript requires proofreading due to multiple errors, to point some of them:
L85: "solubilization"
L101: "synthesizing"
L122: "fertilizers"
L114: "utilized"
L370: missing space
L125, L379: "biofertilizers"
L385: "characterized "
L389: change "routes" to "pathways"
L392: "hightened" to "increased"

---

## Round 0.3 · Minor Revisions

Dear Dr. Mahdavikia and Dr. Dolatabadian,

Thank you for your submission to PeerJ.

It seems that the reviewer is not satisfied with your revisions and insists that you must address the comments before your manuscript is accepted. The reviewer has specifically pointed out some major shortcomings in the discussion section. Accordingly, you are advised to critically revise the discussion section.

If you fail to revise the manuscript critically this time as well, it would be difficult for me to remind you again and your paper will stand rejected.


With regards

·

Basic reporting

pass

Experimental design

pass

Validity of the findings

I would to thank authors for the efforts to improve the manuscript, but it still requires multiple corrections.
Discussion section remained unchanged, it's still lacking the review of previous studies, please highlight the importance of your study. Please avoid common phrases like "Compared to previous studies on Dracocephalum kotschyi, this research has significantly contributed to our understanding of how co-inoculation of AMF and PGPR can enhance the plant's growth and biochemical attributes under water deficit stress. " in the Conclusion.

Additional comments

Manuscript contains some errors even after proofreading:
L31: "calibre" better fits for the weapon than plants
L62: "Fertilisers"
L418: "defence" to "defense"

---

## Round 0.4 · accepted · Accept

Dear Dr. Mahdavikia and Dr. Dolatabadian,

Thank you for your submission to PeerJ.

I am writing to inform you that your manuscript - Co-inoculation of mycorrhizal fungi and plant growth-promoting rhizobacteria improve growth, biochemical and physiological attributes in Dracocephalum kotschyi Boiss. under water deficit stress - has been Accepted for publication.

Congratulations!


This is an editorial acceptance; publication is dependent on authors meeting all journal policies and guidelines.